# An Annular Fresnel Zone Plate without Central Spots Fabricated by Femtosecond Laser Direct Writing

**DOI:** 10.3390/mi13081285

**Published:** 2022-08-10

**Authors:** Xiaoyan Sun, Fang Zhou, Lian Duan

**Affiliations:** State Key Laboratory of High Performance Complex Manufacturing, Central South University, Changsha 410083, China

**Keywords:** Fresnel zone plate, annular structure, femtosecond laser machining, concentric double-ring beams

## Abstract

In recent years, micro-annular beams have been widely used, which has expanded the possibilities for laser processing. However, the current method of generating an annular beam still has shortcomings, such as spot energy at the center of the produced beam. In this study, a Fresnel zone plate with an annular structure was machined using a femtosecond laser. After focusing, an annular laser beam without a spot in the center was obtained, and the radius and focal length of the annular beam could be easily adjusted. In addition, two annular Fresnel zone plates were concentrically connected to obtain a concentric double-ring beam in the same focal plane. The simulation and experimental results were consistent, providing effective potential for applications related to nontraditionally shaped laser beams.

## 1. Introduction

In laser processing, the typical laser parameters include power, spot size, wavelength, pulse duration, and repetition frequency. These parameters are necessary to determine the optimal machining process, and their effects on structure and performance have been extensively studied. The irradiance distribution when the laser reaches the material is also important in material processing. The most common is the Gaussian distribution, which can improve the energy utilization rate to the maximum efficiency. However, in some processes, energy distributions in nontraditional shapes are required. For example, when the diameter of the drilling hole is larger than that of the laser beam [1], the annular distribution of the laser beam can be fired continuously on the material’s surface in the form of a pulse. This method, known as optical perforation, has been successfully used to drill submillimeter holes through stainless steel [2]. In recent years, annular beams have been increasingly used, thereby expanding the possibilities of laser processing [3]. When an annular beam is used together with two-photon polymerization, point-by-point scanning can be optimized to multipoint parallel scanning, and a circular structure can be generated in one process, significantly improving the processing efficiency [4,5]. New imaging techniques can be obtained in stimulated emission depletion microscopes when a circular beam is combined with a Gaussian beam [6,7]. Annular beams can also process microtubules, which have unique applications in biological sensing [8], cell manipulation [9], targeted drug delivery [10], and other fields, attracting the attention of researchers.

Many researchers have studied annular beams from different aspects [11,12,13]. Marti et al. [3] studied the properties, generation methods, and emerging applications of laser beams with nontraditional shapes, such as Bessel beams, annular beams, and vortex beams. They used an annular beam and optical polymerization to generate three-dimensional stacks of circular structures and annular patterns. Arash et al. [14] turned an incident plane wave into an annular beam at its focal plane using a Fresnel zone plate (FZP) with phase shifting radially outward. This method generated an annular beam with high efficiency and good flexibility, but it produced an apparent bright spot at the center of the annular beam. Zhang et al. [15] loaded the phase function of an FZP on a space light modulator to generate an annular beam with variable diameter and realized the rapid processing of tubular structures. They added a high-pass filter to the optical path to filter the light at the center of the annular beam. Fatemeh et al. [16] carried out phase superposition of two radial phase-shifted FZPs to obtain a long-depth bifocal diffraction lens with two focal planes, and a plane with a double-ring beam was obtained between the two focal planes. However, the plane of the double-ring beam was not the focal plane; therefore, energy could not be concentrated when used in laser processing.

In this study, a laser beam with an annular distribution was obtained by focusing the annular Fresnel zone plate (AFZP), and a method of eliminating the center energy of the annular beam was obtained through research. Two AFZPs were concentrically connected, and a focused double-ring beam was obtained on the focal plane. In this study, a simulation calculation was carried out, and an experimental study was carried out by femtosecond laser machining of the AFZP. The results of these two experiments were verified and found to be consistent.

## 2. Theoretical and Experimental Methods

### 2.1. Theoretical Approach

The energy of an annular laser beam is concentrated in the annular region. The different energy distributions in the annular region can be divided into various annular beams, such as the Bezier beam, the optical vortex beam, and the Gaussian distribution of the annular beam. The light field of the Bezier beam is a series of concentric rings with high energy in the center and low energy outside. Therefore, it is necessary to accurately control the processing energy when using a Bezier beam to make the energy density of the inner ring greater than the ablation threshold and the energy density of the outer ring lower than the threshold; otherwise, a concentric annular structure will appear in the processing [17]. The vortex is twisted in a spiral shape around an axis where the light waves cancel each other, making it a circular beam with a radial cross-section. When the topological charge of the vortex light is an integer, the energy distribution of the annular light is uniform in the cross-section. However, when the topological charge is a non-integer, the distribution of the annular light field becomes uneven, which limits the application of vortex light [18].

Researchers have studied methods for generating circular beams. First, an annular beam is produced using an aperture [3]. A Bessel beam can be obtained by irradiating an annular slit on the rear focal plane of the convergent lens with plane waves. However, its efficiency is low because the aperture blocks most of the incoming energy. Annular beams can also be formed using a cone lens [19]. The workpiece is placed sufficiently far away from the cone lens to encounter the far-field profile of the cone lens, which corresponds to a circular beam. However, this method cannot effectively control the size and shape of circular beams. The spatial light modulator (SLM) can change the phase of the incident laser [20,21,22] and generate an arbitrary beam profile and hologram. Therefore, the spatial light modulator can quickly generate an annular beam. However, SLM often has a low damage threshold, which limits micromachining in high-power lasers. In addition, FZP processing on the end face of a multimode fiber is an effective method for generating annular beams [23]. The FZP of the fiber end is conducive to integration into small systems and intrinsically compatible with rapidly developing fiber lasers [24,25,26]. However, optical-fiber FZPs are unsuitable for non-optical-fiber systems because of their narrow application range.

To overcome the shortcomings of Bessel and vortex lights, we used an AFZP to generate a Gaussian circular beam. According to the study of linear FZPs, when the FZP extends indefinitely along the *X-* or *Y*-direction, a grating with varying pitch can be obtained [27]. After the laser passes through the grating, the diffraction pattern in the focal plane is a diffraction line in the same direction as the FZP. A linear FZP can be considered an annular FZP with an infinite radius. When the radius of the annular FZP is reduced to a finite size, an AFZP can be obtained. As shown in Figure 1, the inner radius of the circular FZP is $r_{0}$, and the outer diameter is r. When the number of circular FZP is *n*, $r=\sqrt{n}r_{0}$. Then, radius R of the AFZP is the distance from the center of the AFZP to the center of the circular FZP, the inner diameter is $R_{i}=R-r$, and the outer diameter is $R_{o}=R+r$. According to linear FZP theory, when a laser passes through an AFZP, it will exhibit an annular beam with radius R on the focal plane, and the energy on the focusing ring has a Gaussian distribution. The distance from the focusing ring to the AFZP is $f=r_{0}^{2}/\lambda$, where λ is the wavelength of the incident laser.

A major problem in previous studies on the annular beam is a bright spot in the center of the generated annular beam [14,15]. In ablative machining or material modification, such an annular beam inevitably has a negative effect on the machining morphology, and the center energy should be eliminated. Because the central bright spot is generated during the circular bending of the linear FZP, appropriate structure parameters should be set when designing the AFZP.

Since the AFZP is composed of a transparent and opaque zone, and the area of the opaque zone is larger than that of the transparent zone, when incident light irradiates the AFZP, more than half of the laser light cannot pass through. There is no barrier at the center of the AFZP, and the light beam can pass directly through it. Therefore, incident light passing through the AFZP can be approximated as passing through a circular hole of radius. According to diffraction theory, after the laser passes through the circular hole, it successfully passes through three *Z*-axis zones: the projection area, Fresnel diffraction area, and Fraunhofer diffraction area (as shown in Figure 2). In the projection area of the circular hole, the laser penetrates the circular hole along the *Z*-axis with little energy. In the Fresnel diffraction area, the central spot on the *Z*-axis may be dark or bright, with alternating light and dark spots. In the Fraunhofer diffraction area, a bright spot is always present along the *Z*-axis. The energy at the center of the annular beam is the central bright spot caused by circular-hole Fresnel diffraction or Fraunhofer diffraction.

The approximate range of the Fresnel diffraction area and the Fraunhofer diffraction area produced on the *Z*-axis can be deduced according to the Fresnel diffraction formula. When the radius of the circular hole is *R*_i_, the impulse response function of the light wave field is
(1)$$h\left(x-x_{0},\;y-y_{0}\right)=\frac{e^{jkR_{i}}}{j\lambda R_{i}}.$$

In rectangular coordinates,
(2)$$R_{i}=\sqrt{z^{2}+{\left(x-x_{0}\right)}^{2}+{\left(y-y_{0}\right)}^{2}},$$
where *z* is the distance between the hole and viewing screen. When *z* is large (paraxial condition), the first two terms of the Taylor expansion can be considered.
(3)$$R_{i}\approx z\left[1+\frac{1}{2}{\left(\frac{x-x_{0}}{z}\right)}^{2}+\frac{1}{2}{\left(\frac{y-y_{0}}{z}\right)}^{2}\right]-\frac{{\left[{\left(x-x_{0}\right)}^{2}+{\left(y-y_{0}\right)}^{2}\right]}^{2}}{8z^{3}}+\cdots$$

The essence of the Fresnel approximation is to replace the Huygens wavelet of the sphere with a quadrilateral. Because the phase change caused by the third term of the Taylor expansion is far less than π/2, the Fresnel diffraction area can be deduced as
(4)$$z^{3}\geq \frac{{\left[{\left(x-x_{0}\right)}^{2}+{\left(y-y_{0}\right)}^{2}\right]}^{2}{}_{max}}{2\lambda }=\frac{{\left(2{\left(2R_{i}\right)}^{2}\right)}^{2}}{2\lambda }=32\frac{R_{i}{}^{4}}{\lambda }.$$

Furthermore, Equation (3) can be written as
(5)$$R_{i}\approx z\left[1+\frac{1}{2}{\left(\frac{x-x_{0}}{z}\right)}^{2}+\frac{1}{2}{\left(\frac{y-y_{0}}{z}\right)}^{2}\right]=z+\frac{x^{2}+y^{2}}{2z}-\frac{xx_{0}+yy_{0}}{z}+\frac{x_{0}^{2}+y_{0}^{2}}{2z}.$$

In this formula, the phase change caused by the last term is less than π/2; that is, the phase change caused by different hole positions can be ignored. The Fraunhofer diffraction condition is satisfied when $k\frac{{\left(x_{0}^{2}+y_{0}^{2}\right)}_{max}}{2z}\leq \frac{\pi }{2}$. Because ${\left(x_{0}^{2}+y_{0}^{2}\right)}_{max}=2R_{i}{}^{2}$, the following can be obtained:(6)$$z\geq 4\frac{R_{i}{}^{2}}{\lambda }.$$

The AFZP is composed of many concentric zones; thus, the energy of its *Z*-axis plane is due to the combined superposition of the diffraction of all the circular holes. According to Equations (4) and (6), the positions of the projection area, Fresnel diffraction area, and Fraunhofer diffraction area on the *Z*-axis are positively correlated with the radius of the circular hole.

### 2.2. Experimental Methods

The femtosecond laser has an ultrashort pulse and ultrahigh energy peak [28,29,30], which can realize non-hot-melt cold processing and high-precision processing, and which has a wide range of applications in micro and nano processing [31]. In the experiment, a Yb:KGW femtosecond laser (Pharos, Light Conversion, Vilnius, Lithuania)—with a center wavelength of 1030 nm, pulse duration of 216 fs, and repetition rate of 10 kHz—was used to generate Gaussian beams. The optical path diagram of the femtosecond laser processing system is shown in Figure 3. When exiting from the femtosecond laser provider, it passes through the shutter, attenuator, mirrors, power meter, and apertures, before finally entering the objective lens. The shutter controls the laser switch, the attenuator regulates the light intensity, and two apertures are used to collimate the beam entering the objective [32]. The lens is a 20× objective lens (Nikon, Tokyo, Japan, NA = 0.4). After focusing the laser beam on the vertical incident mode, it is ablated and processed on the surface of the sample. The sample used was fused silica (Gulo company, Luoyang, China, K9). The light transmittance was higher than 90% with high precision, high parallels, and high finish, with a size of 20 × 20 × 0.4 mm. The samples were mounted on a three-dimensional mobile platform (Suruga Seiki Corporation, Tokyo, Japan) with computer-controlled *XYZ* motion in three directions with a movement accuracy of 20 nm.

## 3. Results and Discussion

### 3.1. Study of Single-Ring Annular Fresnel Zone Plate (AFZP)

To study the process of the AFZP generating an annular beam, the AFZP model was established in ZEMAX software. The designed AFZP radius R was 0.6 mm, the inner zone radius $r_{0}$ of the circular FZP was 80 μm, and the zone number of circular FZP n was 29. Therefore, the inner diameter $R_{i}$ of the AFZP was 0.169 mm, and the outer diameter $R_{0}$ was 1.031 mm. The incident light was a flat-topped beam with a wavelength of 800 nm, which was obtained in a focal plane 8 mm from the AFZP, as shown in Figure 4. The radius of the annular beam was 0.6 mm, which is the same as that of the AFZP. The beam on the ring had high energy in the center and low energy on the sides. In addition to the annular beam, a bright spot was observed at the center. It can be seen more clearly in Figure 4b that the energy of the central bright spot was much greater than that of the annular area.

To eliminate the bright spot in the center, appropriate structural parameters need to be selected. For the AFZP with constant *r*, when radius R increases, the radius of each concentric zone increases, and the position of the diffraction area on the *Z*-axis also extends backward. The software modeled AFZPs with different radii to study the influence of the radius R on the central energy of the annular beam.

When the incident laser power was 1 W/mm^2^, the energy in the annular area and center area in the focal plane was as shown in Figure 5. The energy in the annular area remained constant at approximately 10 W/mm^2^, and the energy in the center fluctuated. When R was 0.43 mm-i.e., R=r-the energy of the central bright spot was seven times that of the annular area. When the radius R was 0.8 mm, the central bright spot energy reached a maximum of 58.22 W/mm^2^. However, when R was 1.0 mm, the energy at the center decreased sharply to as low as 0.01 W/mm^2^. These energy fluctuations were caused by Fresnel diffraction, resulting in alternating light and dark in the center. When R expanded to more than 1.6 mm, the central energy remained at a persistently low level of about 0.1 W/mm^2^. According to Equation (4), with the increase in radius R of the AFZP, the position of the Fresnel diffraction area of the circular hole is gradually pushed back on the *Z*-axis, such that it occurs behind the focal plane, causing the focusing ring to appear in the circular hole projection area with no energy in the center. Figure 6 shows the annular beam with a radius R of 1.6 mm, where the focusing ring is clear and there is no bright spot in the center.

In laser processing, it is usually necessary to obtain an annular beam of a specified radius R. Therefore, for an annular beam with an arbitrary radius R, it is crucial to eliminate the central bright spot. As can be seen from the above analysis, the central bright spot can be avoided by focusing the AFZP on the projection area of the circular hole. The position of the focusing ring on the *Z*-axis can be realized by adjusting the radius of the circular FZP $r_{0}$.

Figure 7 shows the focal plane energy distribution of different AFZPs obtained by adjusting $r_{0}$ when the radius R was 0.6 mm. As can be seen from the figure, although $r_{0}$ was different, the energy on the focusing ring did not change significantly, and it was stable at approximately 8 W/mm^2^. Simultaneously, there was a significant change in energy at the center of the *Z*-axis. When the radius of the AFZP $r_{0}$ was 20–30 μm, and the focal plane was within 1.2 mm of the *Z*-axis, there was no energy in the center of the annular beam. This shows that the focal plane was in the circular hole projection area, and there was no circular hole Fresnel diffraction. When $r_{0}$ was 35–80 μm, the focal plane appeared 1.2 mm behind the *Z*-axis, there was energy in the center, and the focal plane was in the Fresnel diffraction area of the circular hole. In summary, a smaller $r_{0}$ resulted in a smaller focal length of the focusing ring and a closer distance between the annular beam and the FZP. However, when $r_{0}$ increased, the focal plane of the annular beam appeared in the Fresnel diffraction area or even in the Fraunhofer diffraction area, which resulted in the enhancement of interference at the center and the appearance of a central bright spot. Therefore, a smaller $r_{0}$ and larger R should be selected to eliminate the bright spot in the center of the circular beam.

The focusing process of an AFZP with $r_{0}$ 80 μm and R 1.60 mm was further studied. Distances of 7.1–8.7 mm before and after the focal plane were selected along the *Z*-axis. The sections presented annular beams without bright spots in the center, but the radii of the annular beams were different. From 7.2 mm to 8.4 mm along the *Z*-axis, the radius increased from 1.45 mm to 1.70 mm. In the 8 mm focal plane of the *Z*-axis, the radius of the annular beam was slightly larger than that of the AFZP, which was 1.61 mm. For a linear FZP whose radius tended to be infinite, the diffraction lines were at the same position before and after the focal plane. However, for an AFZP with a finite radius, the position of the annular beam changed along the *Z*-axis, which was caused by the different energies reaching the focus through the inner and outer zones. As shown in Figure 8, the laser energy brought in by the outer zones was higher than that brought in by the inner zones; thus, the annular beam before and after the focal plane appeared to be scattered [15]. Therefore, the radius of the annular beam could be adjusted by adjusting the position of the *Z*-axis, and the highest energy of the annular beams remained in the focal plane.

The femtosecond laser was applied to micromachine fused silica to verify the theory that the AFZP generates an annular beam. The design of the processing path of AFZP was carried out on the original software. Since the radius and width of each zone were not consistent, the processing path of each zone needed to be designed separately. The zone width was wider than the width of single microgroove processing; thus, it was necessary to carry out ring processing many times. The laser power and speed were kept constant during the processing, and the machining radius was successively increased until the whole zone was prepared. The groove width of single microgroove machining was affected by moving speed and laser power. After careful consideration, the parameters of the femtosecond laser were adjusted to 10 mW and 1 µJ of energy, and the focus size was 3.14 μm. The platform was translated perpendicular to the laser beam at a speed of 1000 μm/s, and the width of the single microgroove machining groove was 8 μm. An AFZP with R = 0.6 mm and $r_{0}$ = 80 μm was fabricated. As shown in Figure 9a, each zone of the FZP was clear and concentric. From a radial point of view, the radii of the different FZPs were consistent with the parameters of the linear FZPs.

The processed sample was placed in the diffraction test light path and irradiated by plane light with a wavelength of 800 nm, and the focal plane pattern was received 8 mm behind the FZP. A clear annular beam is shown in Figure 9b. The radius of the annular beam was equal to the radius R of the AFZP. At the same time, a bright spot can be seen at the center of the circular beam. Figure 9e shows the diffraction pattern when R increased to 1.6 mm. It shows that the center of the annular beam only had the laser passing through the circular hole. These lasers in the center were not focused and had very little energy. Since the theoretical maximum diffraction efficiency of the amplitude-type FZP was only 10.13%, the energy of the beam diffracted on the ring did not significantly exceed that of the center. Therefore, the effect of the energy in the center was negligible when processing with an annular beam.

### 3.2. Study of Double-Ring AFZP

Two annular beams or even multiple annular beams can be obtained by concentrically connecting multiple AFZPs. Figure 10 shows a schematic of the double-ring AFZP. The radius of the inner AFZP is $R_{1}$, the radius of the outer AFZP is $R_{2}$, and the spacing between the two AFZPs is $R_{d}$. Because $R_{2}>R_{1}$, according to Equation (4), the Fresnel diffraction area of the outer AFZP on the *Z*-axis is outside the diffraction zone of the inner AFZP. If the annular beam of the inner AFZP has no central spot, then the outer AFZP would also have no central spot.

A double-ring AFZP with $r_{0}$ = 80 μm, $R_{1}$ = 1.6 mm, and $R_{2}$ = 2.4 mm was simulated. As shown in Figure 11a, two clear concentric rings appeared in the focal plane with no spot at the center, and the spacing $R_{d}\;$ between the two rings was 0.8 mm. For multi-annular beams, the consistency of the energy on different rings is very crucial for machining. As shown in Figure 11c, when $R_{d}$ = 0.8 mm, the peak value of the outer AFZP was 9.8 W/mm^2^, and that of the inner AFZP was 5.3 W/mm^2^, with a significant difference between them. Figure 11b shows that the outer AFZP was moved inward to remove the spacing between the inner and outer FZPs, and $R_{d}$ was reduced to 0.73 mm. The energy of the inner and outer rings was approximately 4 W/mm^2^, and the energy consistency of the two rings was greatly improved.

In practical applications, it is important to obtain a double-ring beam with a specified radius and ring spacing. The spacing $R_{d}$ can be adjusted by adjusting $R_{1}$ and $R_{2}$; however, when the required spacing $R_{d}$ is less than the sum of the widths of the inner and outer AFZPs, i.e., $R_{d}<r_{1}+r_{2}$, the required spacing $R_{d}$ cannot be obtained by adjusting $R_{1}$ and $R_{2}$. At this point, there are two ways to reduce the spacing $R_{d}$, one of which is to reduce $r_{01}$ and $r_{02}$. When $r_{01}$ and $r_{02}$ decrease, $r_{1}$ and $r_{2}$ decrease proportionally, and the spacing $R_{d}$ is significantly reduced. Because multiple focusing rings should be in the same focal plane, $r_{01}$ and $r_{02}$ must be the same when adjusting these parameters. As shown in Figure 11c, when $r_{0}$ was reduced from 80 to 40 μm, $R_{d}$ decreased significantly from 0.73 mm to 0.37 mm. The energy values of the inner and outer rings were similar, both above 4.50 W/mm^2^. In addition, $R_{d}$ can be reduced further by decreasing the zone number n. A smaller zone number n results in smaller values of $r_{1}$ and $r_{2}$. As shown in Figure 11d, to shorten the spacing $R_{d}$, six zones on one side of the AFZP were deleted. After deleting 12 zones of the inner and outer AFZPs, the spacing $R_{d}$ was reduced from 0.37 mm to 0.31 mm. However, the energy on the outer ring decreased to 4.21 W/mm^2^, and the energy on the inner ring decreased to 2.27 W/mm^2^. Therefore, when adjusting the double-ring spacing $R_{d}$, the method of adjusting the inner ring radius $r_{01}$ and $r_{02}$ should be the first choice, which cannot only quickly and substantially adjust $R_{d}$, but also maintain the energy on the ring.

Figure 12a shows the double-ring AFZP fabricated using a femtosecond laser, with $r_{0}$ = 80 μm, $R_{1}$ = 1.60 mm, and $R_{2}$ = 2.33 mm. Figure 12b shows the focusing pattern 8 mm behind the AFZP. Two clear focusing rings can be observed. The two focusing rings had high concentricity and appeared simultaneously in the same focal plane, and there was no bright spot in the center. More focusing rings can be obtained in the focal plane if the same $r_{0}$ AFZP is sheathed on the outer side of the double-ring AFZP. Moreover, the distance between different AFZPs $R_{d}$ can be easily adjusted using the forementioned method.

## 4. Conclusions

The AFZP designed in this study not only eliminates the central energy of the single annular beam, but it also obtains a double-ring beam with a controllable radius in the same focal plane. In addition, the AFZP can be easily integrated into the radial direction to obtain a multi-ring beam, and it can also be arrayed in a plane to obtain an annular beam array. Further research on annular beams will provide a more productive potential for applications related to nontraditionally shaped laser beams.

## Figures and Tables

**Figure 1 micromachines-13-01285-f001:**
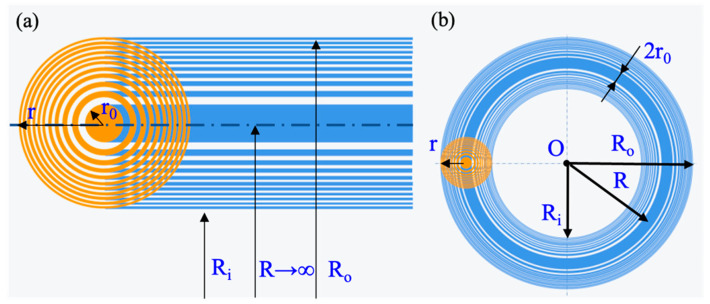
Schematic diagram of Fresnel zone plate (FZP) structure: (**a**) linear FZP; (**b**) AFZP.

**Figure 2 micromachines-13-01285-f002:**
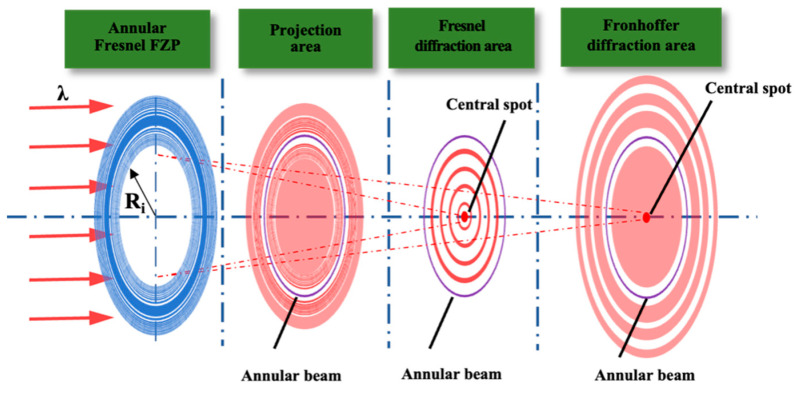
After the laser passes through the circular hole of the AFZP, it successively passes through the projection area, Fresnel diffraction area, and Fraunhofer diffraction area.

**Figure 3 micromachines-13-01285-f003:**
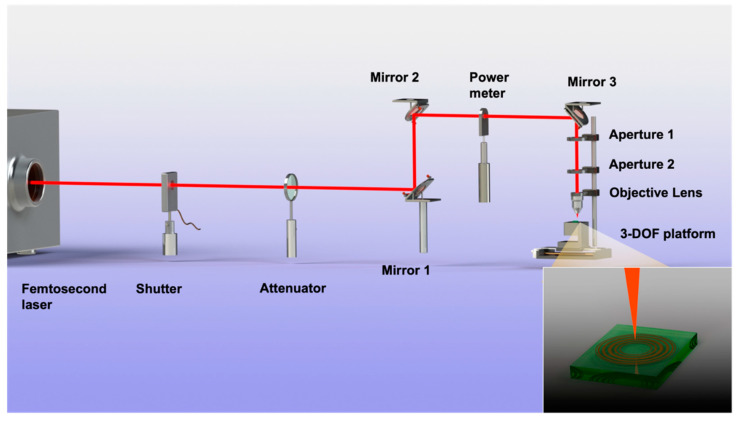
Schematic diagram of femtosecond laser processing system.

**Figure 4 micromachines-13-01285-f004:**
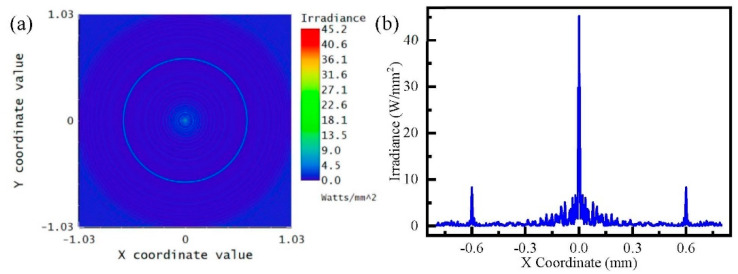
Simulation of annular Fresnel zone plate (AFZP) with R as 0.6 mm and $r_{0}\;$ as 80 μm: (**a**) annular beam and central spot in focal plane; (**b**) energy distribution in the focal plane.

**Figure 5 micromachines-13-01285-f005:**
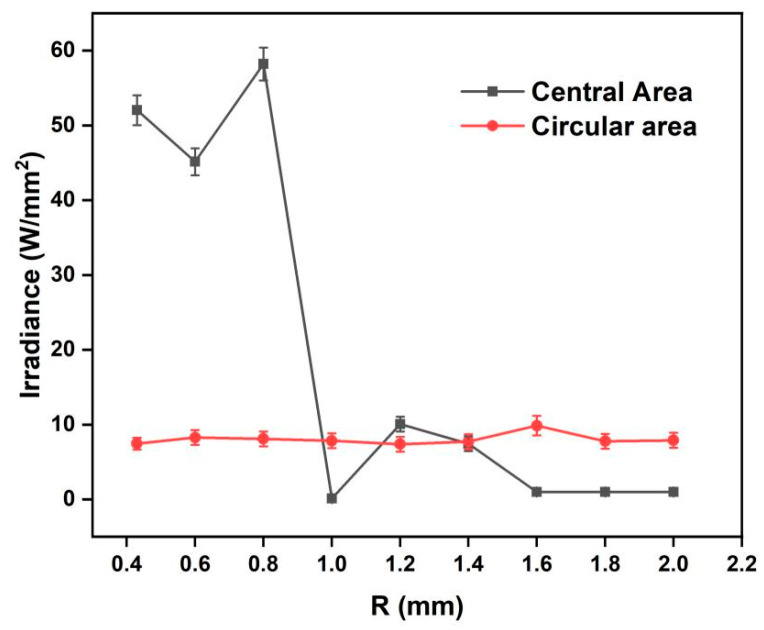
Simulation of the energy on the annular and central areas of the focal plane of AFZPs with different radii R.

**Figure 6 micromachines-13-01285-f006:**
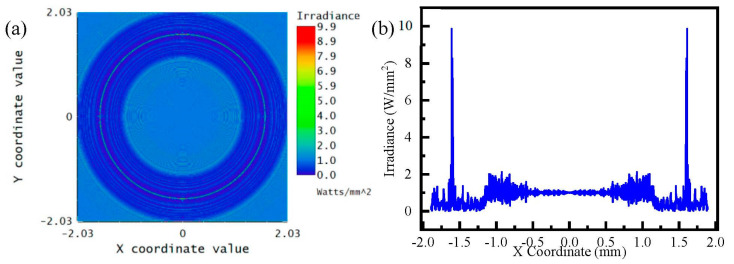
Simulation of AFZP with R as 1.6 mm and $r_{0}$ as 80 μm: (**a**) annular beam in the focal plane; (**b**) energy distribution in the focal plane.

**Figure 7 micromachines-13-01285-f007:**
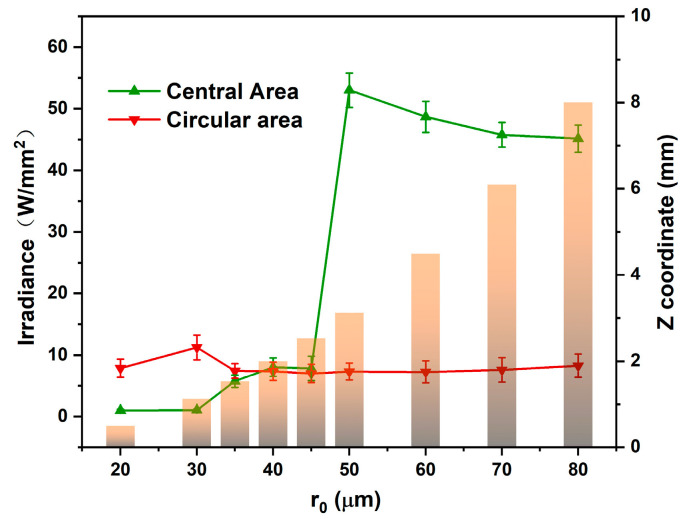
Simulation of the energy on the annular and central areas of the focal plane of AFZPs with different radii $r_{0}$.

**Figure 8 micromachines-13-01285-f008:**
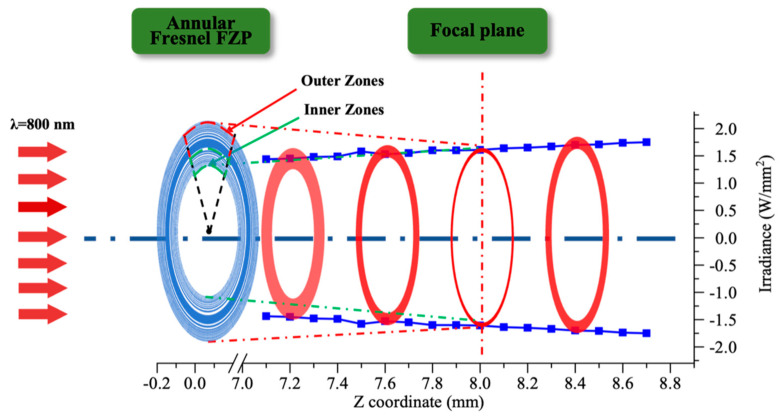
Focusing process of the annular beam before and after focal plane.

**Figure 9 micromachines-13-01285-f009:**
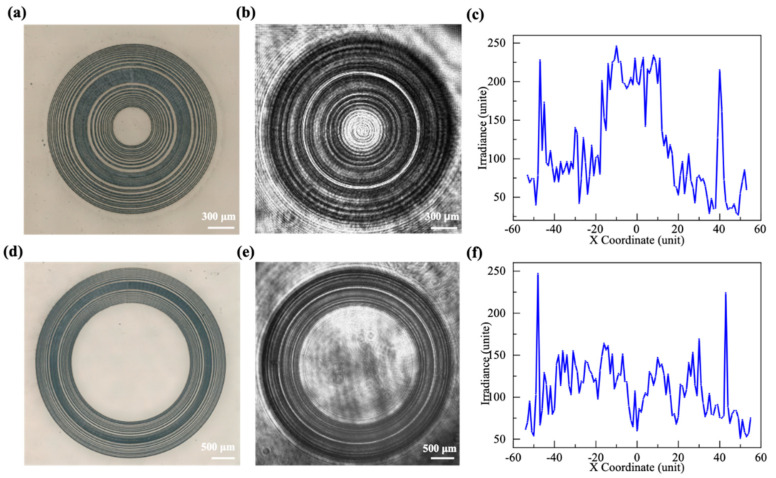
AFZP with R of 0.6 mm with central spots: (**a**) the AFZP prepared by a femtosecond laser on a fused silica sample; (**b**) the focusing ring behind the AFZP photographed by a CCD; (**c**) the energy distribution diagram of the focal plane, for the AFZP with R = 1.6 mm without central spots; (**d**) the AFZP prepared by a femtosecond laser on a fused silica sample; (**e**) the focusing ring behind the AFZP photographed by a CCD; (**f**) the energy distribution diagram of the focal plane.

**Figure 10 micromachines-13-01285-f010:**
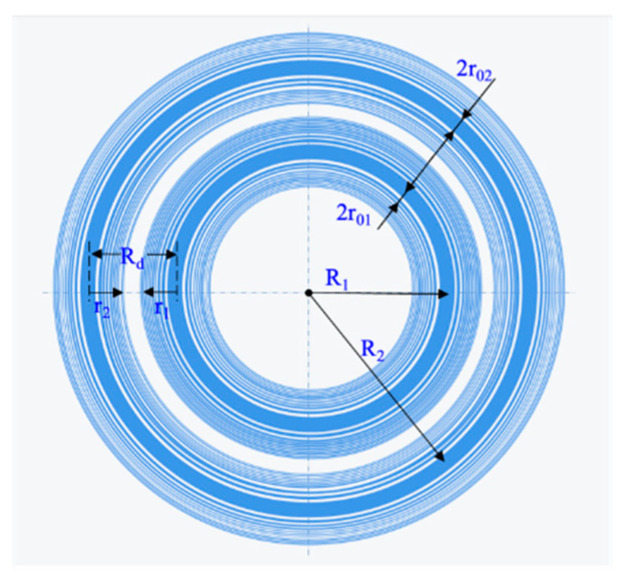
Schematic diagram of double-ring AFZP.

**Figure 11 micromachines-13-01285-f011:**
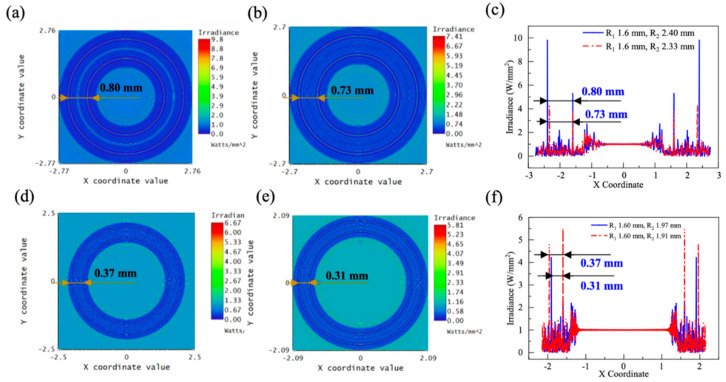
Simulation pattern of double-ring AFZP with $R_{1}$ of 1.6 mm: (**a**) $R_{2}$ = 2.40 mm, $R_{d}$ = 0.80 mm; (**b**) $R_{2}$ = 2.33 mm, $R_{d}$ = 0.73 mm; (**c**) comparison of energy distribution in a/b diagram; (**d**) $R_{2}$ = 1.97 mm, $R_{d}$ = 0.37 mm; (**e**) deletion of 12 zones, with $R_{2}$ = 1.91 mm, $R_{d}$ = 0.31 mm; (**f**) comparison diagram of energy distribution of d/e diagram.

**Figure 12 micromachines-13-01285-f012:**
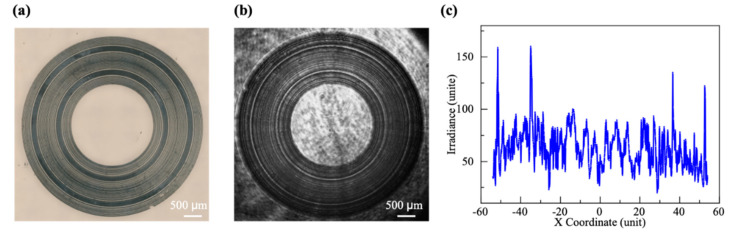
(**a**) The double-ring AFZP with $R_{1}$ = 1.60 mm and $R_{2}$ = 2.33 mm was fabricated using a femtosecond laser on a fused silica sample; (**b**) focusing double-ring behind the AFZP captured by a CCD; (**c**) energy distribution diagram of the focal plane.

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
