# Peer review of "An Annular Fresnel Zone Plate without Central Spots Fabricated by Femtosecond Laser Direct Writing"

_micromachines, 2022, doi:10.3390/mi13081285_

Round 1

Reviewer 1 Report

This is an interesting paper which can be publish after minot revisions and following mandatory comments belown. Note this is important to really do these improvements.

First, I think there is strong need to cite teh first papers on femtoseconde laser writing of Fresnel lens in similar materials namely: 

·      In 2002 : https://doi.org/10.1364/OL.27.002200

And https://doi.org/10.1364/OE.10.000978

·      In 2004 ; Yamada, K.; Watanabe, W.; Li, Y.; Itoh, K.; Nishii, J. Multilevel Phase-Type Diffractive Lenses in Silica Glass Induced by Filamentation of Femtosecond Laser Pulses. Opt. Lett. 2004, 29, 1846.

Then you have to mention very recent work and progress in writing such Fresnel lens by femtosecond laser in glasses and extending application, range in the mid-IR:

·      on chalcogenide : https://www.mdpi.com/2076-3417/12/9/4490/htm

·      on heavy oxyde glass : http://dx.doi.org/10.2139/ssrn.4110711

Need to add some error bars in Figs 5 and 7 as much as possible (even I know this is some mdeoling!)

Missing details about writing process in experimental section and in the printing (page 11, line 293):

·      About sample composition: it is said silicon dioxide and some sometimes silica (so it is likely Fused silica ?) but should rather give more details like providers and silica name and ideally its type (like suprasil, infrasil, from Heraeus or silica from Corning, Schott etc..

·      Laser (1030, 10kHz), laser provider (name, model) and energy (to put in microJoules and not only in mW) parameters, speed (looks 1000microns/s),  numerical aperture of the x20 objective,  focusing depth ?, pulse duration ?

·      Estimated beam spot size that it important when speaking about the written geometry and its resolution

·      Then need to explain the writing design and its trajectory (spirals trajectory? Concentric rings ?  spirals or rings increment in microns comparing to the beam size ?, only single layer writing ?)

What was the total writing time for these « microlenses » .?

·      What is the targeted phase shift (in radians) at which wavelength? How dod you choose the energy and speed to effecctively reach to desited phase changes at the design wavelength ?

·      What is the effectively implemented quantitative phase in radians and at which wavelength ? did you measured it ?

Images 11 and 12: the contrast if not always so good and this needs so add some comments about it. It is could related to the imaging quality but also she might indicate a lack of efficiency of these Fresnel lenses because 1/ some deviation of the laser imprinted phase profiles compared to the target value of 2.Pi radians for example or on the design which can be improved with a multilayers writing approach (e.g. 8 layers used in Ref. https://www.mdpi.com/2076-3417/12/9/4490/htm).

Author Response

Dear reviewer and editor:

Thank you for commenting on our article. After carefully revise and promotion, the quality of manuscript was improved significantly. Please see the attachment.

Kind regards, 

Dr. Duan

Reviewer 2 Report

The manuscript is devoted to the study important problem namely designing and machining controllable beam shaping an annular laser beam by a femtosecond laser laser and Fresnel zone plate. Authors proposed and experimentally verified the technique for obtaining annular laser beam without a spot in the center. In addition, two annular Fresnel zone plates were connected to obtain a concentric double-ring beam in the same focal plane.  The results of numerical simulations are well illustrated. The manuscript is well-structured. The advantage of the manuscript is that some theoretically predicted results are proved by the conducted experiment.

But there is the following remark:

Line 42. The reference “Duocastella [3] studied…” is incorrect.

Line 75. The expression “…the number of laser torsions in wavelength is an integer…” is incorrect.

Line 80. Reference [3] is indicated incorrect.

Line 171. “Simulation” is to be deleted.

As is seen from Figs. 11 and 12 describing focusing ring behind AFZP the laser power outside the annular field is essentially larger than the power contained in ring. Here the intensity in  on-axial area amounts to approximately 60 percent from the intensity in ring. That is why spot energy at center is not excluded that contradicts to the main result of the paper. In this connection the question arises on reasonability of using (compared to AFZP) an optical scheme on the basis of an axicon and a lens for formation of annular fields with zero intensity in by-axial area. That is why it is necessary to explain why the AFZP scheme proposed by the authors is better that the scheme on the basis of an axicon and a lens. 

In the text of the manuscript there are a lot of places of wrong usage of prepositions.

English should be considerably improved. A special attention is to be paid to the usage of prepositions in the whole text of the manuscript.

After corrections, I do recommend to publish this manuscript in Micromachines.

Author Response

Dear reviewer and editor:

Thank you for commenting our article. After carefully revise and promotion, the quality of our manuscript has been improved significantly. Please see the attachment.

Kind regards,

Dr. Duan

This manuscript is a resubmission of an earlier submission. The following is a list of the peer review reports and author responses from that submission.

Round 1

Reviewer 1 Report

In the manuscript, the authors investigate the annular Fresnel zone plate (AFZP). Linear FZP and AFZP are introduced and different parameter effects are simulated. To prove the simulated results, AFZP structures are fabricated by using a femtosecond laser machining. The measured energy distributions on the focal plane show similar patterns obtained from the simulation. I cannot recommend its publication in its current version. The detailed comments are the following.

(1)  The logic structure of the manuscript should be improved. For example, the title of Section 2.2 is “Theoretical and experimental methods”. However, in this section, there is nothing related to the theory. A large part of text in this section is about the summary of the previous studies, which is, obviously, should not be included in this section. Furthermore, in this study, two parameters, i.e., R and r0, are important for the design of AFZPs. I think the authors should first do the parameter studies, e.g., Figs. 5 and 7, and then show the detailed results for some parameters.

(2)  I suggest the authors to revise the manuscript title to clearly state the main idea of the work.

(3)  Figure 1: what is the difference between the parameter r in (a) and (b)? From the text, it looks r in the two cases are identical. However, from the drawings in Figure 1, the two r are completely different. The authors should explain this.

(4)  When the authors explain the experimental setup (Fig. 2), the laser central wavelength is 1030 nm. However, in line 125, the authors state that the incident light is with a wavelength of 800 nm. Why aren’t the wavelengths same?

(5)  The authors should explain how they choose the parameters R and r0.

(6)  Line 128: 80 um is for r0 rather than R0.

Reviewer 2 Report

This manuscript doesn't fit into the scope of this journal.

Reviewer 3 Report

In the manuscript, the authors report an annular Fresnel zone plate that produces a micro-annular beam with no spot in the center by femtosecond laser machining. The radius and focal length of the annular beam could be easily adjusted. Besides, a concentric double-ring beam in the same focal plane is obtained by concentrically connecting two annular Fresnel zone plates. The experiment results match well with the simulation.

From my perspective, this study is important for some specific applications of annular beams. The manuscript is well-organized and clearly presented in general. I would recommend accepting the manuscript but with minor revisions. The concerns are listed as follows.

1.       As described in Page 9 Row 280, there should be another figure 11 which is missing in the manuscript.

2.       The writing should be improved dramatically. There are a lot of typo mistakes.

3.       Some figures need to be modified (e.g. different font size in one figure, sometimes illegible font size, no units in axis label, etc.). The authors should strictly follow the style format when plotting. 

Reviewer 4 Report

I have read with great interest the manuscript by Xiaoyan et al., “Femtosecond laser machining of annular Fresnel zone plate produces a micro-annular beam with no spot in the center.” The authors report a typical annular Fresnel zone plate (AFZP) which was machined using a femtosecond laser and a method of eliminating the center energy of the annular beam was obtained. I could see a major advancement of this work: Two AFZPs were concentrically connected, and a focused double-ring beam was obtained on the focal plane. The simulations and experiments are reasonably conducted. Nevertheless, I do have a major concern on how could current manuscript be related to Sensors journal without relevant application. I believe this work could be of interest in the domain of laser material processing. Therefore, I feel it could be suitable for publication in relevant journals like Micromachine, Optics journals.

Few Comments:

1.     Authors should provide descriptions of the experiments in enough detail so that a normal researcher is able to understand and repeat them

2.     Authors mentioned energy in many places where Intensity or irradiance needs to be mentioned